# Prediction of Early Response to Immunotherapy: DCE-US as a New Biomarker

**DOI:** 10.3390/cancers14051337

**Published:** 2022-03-04

**Authors:** Raphael Naccache, Younes Belkouchi, Littisha Lawrance, Baya Benatsou, Joya Hadchiti, Paul-Henry Cournede, Samy Ammari, Hugues Talbot, Nathalie Lassau

**Affiliations:** 1Department of Imaging, Institut Gustave Roussy, 94800 Villejuif, France; baya.benatsou@gustaveroussy.fr (B.B.); joya.hadchiti@gustaveroussy.fr (J.H.); samy.ammari@gustaveroussy.fr (S.A.); nathalie.lassau@gustaveroussy.fr (N.L.); 2CVN INRIA, CentraleSupelec, Universite Paris-Saclay, 91190 Gif-Sur-Yvette, France; hugues.talbot@centralesupelec.fr; 3Laboratoire d’Imagerie Biomédicale Multimodale Paris-Saclay, BIOMAPS, UMR 1281, Université Paris-Saclay, Inserm, CNRS, CEA, 94800 Villejuif, France; littisha.lawrance@gustaveroussy.fr; 4MICS Lab, CentraleSupelec, Universite Paris-Saclay, 91190 Gif-Sur-Yvette, France; paul-henry.cournede@centralesupelec.fr

**Keywords:** DCE-US, perfusion, biomarker, immunotherapy, overall survival

## Abstract

**Simple Summary:**

Immune checkpoint inhibitors (ICI) have revolutionized cancer care. However, assessing the efficacy of these new molecules with targeted therapeutic responses may induce too much delay when using classical biomarkers derived from morphological imaging (CT). The objective of our study is to propose fast, cost-effective, convenient, and effective biomarkers using the perfusion parameters from dynamic contrast-enhanced ultrasound (DCE-US) for the evaluation of ICI early response. In a population of 63 patients with metastatic cancer eligible for immunotherapy, we demonstrate that a decrease of more than 45% in the area under the perfusion curve (AUC) between baseline and day 21 is significantly associated with better overall survival. Thus, AUC from DCE-US looks to be a promising new biomarker for the early evaluation of response to immunotherapy.

**Abstract:**

Purpose: The objective of our study is to propose fast, cost-effective, convenient, and effective biomarkers using the perfusion parameters from dynamic contrast-enhanced ultrasound (DCE-US) for the evaluation of immune checkpoint inhibitors (ICI) early response. Methods: The retrospective cohort used in this study included 63 patients with metastatic cancer eligible for immunotherapy. DCE-US was performed at baseline, day 8 (D8), and day 21 (D21) after treatment onset. A tumor perfusion curve was modeled on these three dates, and change in the seven perfusion parameters was measured between baseline, D8, and D21. These perfusion parameters were studied to show the impact of their variation on the overall survival (OS). Results: After the removal of missing or suboptimal DCE-US, the Baseline-D8, the Baseline-D21, and the D8-D21 groups included 37, 53, and 33 patients, respectively. A decrease of more than 45% in the area under the perfusion curve (AUC) between baseline and D21 was significantly associated with better OS (*p* = 0.0114). A decrease of any amount in the AUC between D8 and D21 was also significantly associated with better OS (*p* = 0.0370). Conclusion: AUC from DCE-US looks to be a promising new biomarker for fast, effective, and convenient immunotherapy response evaluation.

## 1. Introduction

Immune checkpoint inhibitor (ICI) immunotherapies using monoclonal antibodies antagonizing T-cell co-inhibition receptors have been the major revolution of the last few years in anti-cancer treatment. By blocking the immune checkpoints used by the tumor cells to create an immunosuppressive tumor microenvironment, ICIs enhance the antitumor immune response [1]. Since the first FDA approval of an ICI (ipilimumab for the treatment of advanced melanoma both in pre-treated or chemotherapy naïve patients, in March 2011 [2]), many more therapeutic extensions and molecules have been approved [3]. The effectiveness of ICIs in metastatic cancer is no longer a question in terms of survival gain or sustainable response compared to chemotherapy. However, except for melanoma and Hodgkin lymphoma, which show an excellent response rate (>50%) [4,5], only a subset of patients exhibits a good response with immunotherapy. We cite the following as examples: for advanced-stage hepatocellular carcinoma, the response rates for nivolumab and pembrolizumab were respectively 19.7% and 20.7% [6]; in advanced-stage small cell lung cancer, characterized by its aggressiveness and early diffusion of metastases, the rate of response for pembrolizumab in a phase II trial was 19.3% [7]; for advanced squamous-cell non-small lung cancer, the rate of response for nivolumab in a phase III study was 20% [8]; for metastatic DNA mismatch repair-deficient/microsatellite instability-high colorectal cancer, which also has a poor prognosis following conventional chemotherapy, the rate of response for nivolumab was 31.1% in a phase II trial [9].

This highlights the need to rapidly evaluate the efficacy of immunotherapy to avoid wasting valuable time and resources since the majority of patients will not respond to these expensive molecules. Furthermore, the use of these new molecules has been associated with unconventional response patterns, such as pseudo-progression, which is defined as an objective response following initial progression with the same treatment. Pseudo-response has been reported with an incidence rate of up to 10% [10]. To deal with these new patterns of responses, the usual criteria for evaluating chemotherapy response, the Response Evaluation Criteria in Solid Tumors 1.1 (RECIST 1.1), was updated with immune RECIST1.1 (iRECIST 1.1) [11,12]. The differences are small but essential. The progression category includes two new subcategories. The first is immune Unconfirmed Progressive Disease (iUPD), which is labeled a progression according to RECIST 1.1. However, with iRESIST 1.1, a progression needs to be confirmed 4 to 8 weeks later by a new increase in lesion size to be included in the second subcategory of progressive disease, namely, immune Confirmed Progressive Disease (iCPD). This increases the delay before declaring a patient as a non-responder and, therefore, the delay in changing the therapeutic line. Hence, morphological images, which form the basis of iRESIST 1.1, are not useful for predicting early response to ICIs. In summary, ICIs are a great alternative to minimize the low efficacy of chemotherapy for some advanced cancers with a greater overall survival rate. Above all, they offer a sustainable response for patients that respond well. However, the response rate is only 20–30% when considering all cancer types together. Thus, early response evaluation has become a major requirement to stop ineffective treatments earlier, which is not possible at present with the assessment from iRECIST 1.1 CT-scans.

To the best of our knowledge, there are currently no studies in the literature that have looked at dynamic contrast-enhanced ultrasound (DCE-US) for the purpose mentioned above. DCE-US is a real-time functional imaging modality with high temporal resolution and sensitivity to contrast agents. It highlights the signal from a microbubble-based contrast agent within the tumor micro-vascularization, making it possible to follow the tumor vascularization over time using a time-intensity curve (TIC) [13]. It has already been shown to be effective in the early response assessment to antiangiogenic drugs [14]. This technique brings to light the inflammation process induced by ICI and observes the tumor destruction by the immune system. After ICI administration, a lymphoproliferation is observed, resulting in an influx of immune cells [15], leading to an increase of perfusion followed by a decrease due to necrosis induced by the destruction of tumor cells and vasculature. Our study aims to determine whether perfusion parameters extracted from dynamic contrast-enhanced ultrasound can be used as biomarkers for ICI early response evaluation.

## 2. Materials and Methods

### 2.1. Patients

This retrospective study enrolled 63 patients with metastatic melanoma, colorectal cancer, pulmonary cancer, kidney cancer, liver cancer, cervical uterus cancer, or sarcoma, eligible for immunotherapy treatment (atezolizumab, nivolumab, or pembrolizumab). All these patients were included from three phase I or IIB clinical trials to assess the efficacy of the combination of systemic ICI and local treatment in patients with metastatic tumors. The inclusion criteria were as follows: (a) patients with metastatic cancer, (b) treatment with ICIs used alone or in association with other modalities of treatment, (c) patients older than 18 years of age, (d) target lesion accessible by ultrasound, and (e) tumor size larger than 10 mm at baseline in B mode. The exclusion criteria for this study were all the contraindications to the use of the sulfur hexafluoride (Sonovue^®^): hypersensitivity to the sulfur hexafluoride, uncontrolled systemic hypertension, severe pulmonary arterial hypertension, recent acute coronary syndrome, unstable ischemic heart disease, right–left shunt, respiratory distress syndrome, as well as pregnant women or breast-feeding women. All patients signed informed consent forms.

### 2.2. DCE-US Technique and Quantification

The DCE-US examinations were conducted using an Aplio™ 500 ultrasound system (Canon, Puteaux, France). Depending on the metastatic site, two different probes (3.5 and 8 MHz) were used. The Aplio™ ultrasound system had access to the raw linear data using Vascular Recognition Imaging (VRI), a perfusion software, and CHI-Q quantification software, as in a previous study [14]. Standardized procedures were performed: first, a morphological analysis was undertaken to determine tumors sizes in all three dimensions with electronic calipers; then, the perfusion study was conducted after an intravenous bolus injection of 4.8 mL of Sonovue^®^ (Bracco, S.P.A., Milan, Italy), followed by the perfusion of 5 mL of physiological serum. The perfusion curve was recorded for 3 min immediately after the Sonovue bolus. The time-intensity curve of the tumor perfusion was then modeled by a DCE-US study leader using a mathematical model based on the indicator-dilution theory that models the flow of contrast microbubbles in the vasculature [13] and the software already mentioned.

Seven perfusion parameters were then measured, four of which are related to blood volume (peak intensity, area under the curve (AUC), AUC during the wash in, AUC during the wash-out); two to blood flow (time to peak intensity, slope of the wash in); and the last parameter to the mean transit time. The parameters are represented in Figure 1.

### 2.3. Assessments

The DCE-US examinations were performed at baseline, day 8 (D8), and day 21 (D21) after the beginning of the treatment. For this study, the chosen target lesion was treated exclusively with ICIs without any additional local treatment. The tumor perfusion curve was modeled on these three dates to produce the seven DCE-US perfusion criteria described previously. The change in perfusion parameters was then measured between baseline-D8 and baseline-D21. In order to study the variation of perfusion due to the administration of ICI, we focused on the increase of tumor perfusion at D8 and its decrease at D21.

### 2.4. Analysis

The base of our evaluation was overall survival (OS), defined as the time between the first DCE-US at baseline and death from any cause. Median and interquartile range (IQR) were used to report the distribution of the variation of perfusion parameters. The Kaplan–Meier method was used for univariate analyses and Cox regression for multivariate analyses. DCE-US variation parameters were then used to separate the population into two subgroups to show its impact on OS. The log-rank test was used to compare the survival distribution of the categories obtained from the univariate parameters and compute the *p*-value to assess the significance of the comparison. The perfusion parameters in each survival group were then reported in box plots, and a Mann–Whitney U-test of independence was performed to show the significance of the difference in parameters. The thresholds used to evaluate the chosen criteria were computed using maximally selected rank statistics [16,17].

The statistical analyses were performed using Python version 3.8, with the lifelines package v0.25.7, pandas v1.1.5, statannotations v0.4.2, and scipy v1.6.3.

## 3. Results

### 3.1. Population

Between November 2016 and February 2021, 63 patients were enrolled in this study. Patient baseline characteristics are listed in Table 1.

Among the 63 patients initially included, 25 patients had no ultrasound at D8 in their study protocol, 1 patient had a D8 DCE-US with unsatisfactory quality, and 10 had no ultrasound quantification at D21 (6 suboptimal qualities and 4 ultrasounds not performed). Thus, there were 37 remaining patients in the D8 group and 53 in the D21 group (Figure 2).

### 3.2. At D8

The analyses of changes from baseline to D8 revealed no significant association with OS for any of the perfusion parameters (Table 2). However, the difference in the AUC_wash in_ at D8 showed the strongest association (*p* = 0.0592).

A maximally selected ranking test with a constraint of a minimum of 30% population in each class determined the best cutoff point for this parameter: an increase in AUC_wash in_ between baseline and D8 greater than 20% would seem to be associated with better OS (*p* = 0.3111) (Figure 3). This threshold of 20% separated the 37 patients into two groups: the first composed of 12 patients with good overall survival (median OS not reached), the second consisting of 25 patients with poor overall survival.

The variation of the perfusion criteria from baseline to D8 in these two groups is represented in a box plot (Figure 4). It shows an increase in all perfusion parameters in the better overall survival group, with a significant difference in most parameters.

### 3.3. At D21

The analyses of changes from baseline to D21 revealed a significant association with OS for most of the perfusion parameters (Table 3). Change in the AUC at D21 was the most important criterion, showing the strongest association with OS (*p* = 0.0028).

A maximally selected ranking test with a constraint of a minimum of 30% of patients in each class on the variation of the AUC brought out a cutoff point at 45%: a decrease greater than 45% in AUC between baseline and D21 was significantly associated with better OS (*p* = 0.0114) (Figure 5). This threshold at 45% separated the 53 patients into two groups: the first was composed of 20 patients with good overall survival (median survival not reached at the endpoint date), and the second consisted of 33 patients with poor overall survival.

The variation of the perfusion criteria from baseline to D21 in these two groups is represented in the box plot (Figure 6). It shows a significant decrease for most perfusion parameters in the better overall survival group.

A Cox regression model was fitted on the dataset to show the effect of age and gender as covariates on survival. Decrease by 45% of the AUC was the only parameter significantly correlated to survival (HR = 1.75, *p* = 0.05). The results of the Cox regression are summarized in Figure 7.

### 3.4. Change in AUC between D8 and D21

For the most significant parameter, its variations amongst all three-time points were studied for the eligible patients: 33 patients had a DCE-US at D8 and D21. In the group with better survival at D21 (corresponding to the group of patients with a decrease of more than 45% in AUC compared to baseline), analyses of the changes in AUC between D8 and D21 revealed in all patients (without exception) a decrease in AUC between these two dates. Conversely, in the group with poorer survival, all patients showed an increase in AUC between D8 and D21 (except for two patients) (Figure 8). Thus, the decrease in AUC between D8 and D21 is significantly associated with better overall survival (*p* = 0.0370) (Figure 9).

## 4. Discussion

With the advent of immunotherapy, revolutionizing therapeutic cancer management, ICIs are being studied in numerous cancers. With the significant increase in the use of ICIs, atypical new response patterns have emerged, such as pseudo-progression. It is defined as an increase in the size of lesions or the appearance of new lesions, followed by a potentially long-lasting positive response. Therefore, it is necessary to develop new methods to assess the efficacy of ICIs. In clinical routine, iRECIST 1.1 is accepted as the new reference in ICI scan evaluation. The guidelines for response criteria of iRECIST 1.1 are well described in an article published in The Lancet Oncology in 2018 [11]. However, these criteria are based on morphological analysis, and they increase the delay by 1 month compared to RECIST 1.1 before being able to associate patients with a progressive disease, which might be critical. Thus, there is an important need to find a tool for the early assessment of ICI response. With this view, we investigated whether DCE-US, with the study of perfusion parameters, could be useful for the early assessment of the therapeutic response.

Our study confirmed that the early evaluation of AUC at D21 may be used to predict survival after treatment with ICIs. Indeed, the decrease of AUC by more than 45% at D21 is associated with better overall survival (*p* = 0.01). To our knowledge, this is the first study evaluating DCE-US in patients with metastatic cancer treated with ICIs. Our results are consistent with a large multicenter cohort study published in 2014, which confirmed that DCE-US could be used to predict early progression and overall survival after antiangiogenic therapy in metastatic cancers [14]. In this study, AUC was also found to be the best performing criterion for this evaluation, with a decrease in AUC at 1 month of more than 40% associated with better overall survival (*p* = 0.05). Although the cutoffs are not identical, our study was highly motivated by that result since the studied criterion (decrease of AUC) as a relevant marker for overall survival is the same. Moreover, in a study assessing the reproducibility of DCE-US perfusion parameters, AUC was found to be the most robust criterion [18].

Furthermore, our results at D8 (although not significant) and D21 show a trend. It appears that in patients with prolonged overall survival, perfusion increases at D8 and decreases strongly at D21. This result seems to be consistent with the changes in the tissue level induced by the introduction of ICIs. Immune checkpoint blocking is shown to activate and lead to the proliferation of T-cells and NK-cells [15,19]. A pro-inflammatory environment is created. One of the aspects of this environment is the initiation of neo-angiogenesis that allows an influx of immune cells (in particular CD8+ LTs) to eliminate the tumor cells, leading to the destruction of tumor vasculature along with tumor cells. This results in necrosis, which explains the decrease of perfusion in case of response. This phenomenon may explain why an increase in perfusion at D8 in the tumor site would indicate a good response to immunotherapy (witnessing the influx of immune cells), while a decrease in perfusion at D21 would also indicate a good response (witnessing the tumor necrosis).

Currently, more and more studies are looking at non-morphological criteria to evaluate the response to ICIs. The metabolic response of tumors also seems to be a good way to assess ICIs. In a retrospective study with a small cohort of 28 patients with non-small cell lung cancer (NSCLC) treated with nivolumab, authors evaluated the potential of FDG PET/CT to monitor ICIs’ response [20]: they used a modified PERCIST (PET Response Criteria in Solid Tumors), iPERCIST (immune PET Response Criteria in Solid Tumors), which is a dual-time-point evaluation of “unconfirmed progressive metabolic disease” status after the first PET scan evaluation, followed by a new evaluation after 4 weeks to confirm or deny a progressive metabolic disease (which is similar to iRECIST). In the study, iPERCIST is a good tool to separate responder and non-responder patients, with significantly better overall survival in responder patients (*p* = 0.0003). Moreover, the comparison of iPERCIST with iRECIST showed a reclassification in 39% of the 28 patients with relevant additional prognostics. However, we are confronted with the existence of a delay, as with iRECIST, before being able to declare patients as non-responders. In another prospective study with a small cohort of 24 patients, authors investigated whether ^18^F-FDG-PET/CT could predict the therapeutic response of an ICI (nivolumab in NSCLC) in the early phase [21]. They showed that at one month, the ^18^F-FDG uptake with the measure of total lesion glycoses (TLG) could significantly predict partial response (*p* = 0.021) and progressive disease (*p* = 0.002). Furthermore, a statistically significant difference in the predictive probability of response was found between TLG by PET and CT scans at 1 month (*p* = 0.0007).

The perfusion was also analyzed to assess the response of ICIs with DCE-MRI, particularly in a study assessing DCE-MRI perfusion to predict pseudo-progression in metastatic melanoma treated with immunotherapy [22]. With a cohort of 44 patients, the authors highlighted that the plasma volume (Vp) was significantly lower in pseudo-progression than in real progression (*p* = 0.04).

These studies represent interesting perspectives to complement morphological analysis in the early evaluation of the response to immunotherapy. However, unlike these tools, DCE-US is much less expensive, less invasive, much more readily available, with the possibility of repeating these examinations regularly after the onset of treatment to study the perfusion profile. There is also no contraindication against renal failures, which is not the case for the other techniques mentioned above. They are also much less convenient and are associated with harmful effects if repeated too frequently. These effects include irradiation due to PET/CT or the accumulation of gadolinium in the brain, for which we do not have the necessary hindsight concerning the safety of repeated injections. In addition to the previously mentioned tools, there is a new rising approach using artificial intelligence (AI) and radiomics models as predictive biomarkers of response to ICIs [23]. Trebeschi et al. [24] performed an AI-based characterization of 1055 lesions from 203 patients with advanced melanoma and NSCLC undergoing anti-PD1 therapy on the pretreatment contrast-enhanced CT imaging data. In this study, significant performances were observed to predict OS for both tumor types (*p* < 0.001) with ICIs. However, the predictive performance varied depending on the site of the metastases, with non-significant performance for liver metastases (*p* = 0.13) or adrenal metastases (*p* = 0.18). Khorrami and al. [25] developed a radiomics model based on changes in pretreatment and early post-treatment (6–8 weeks) CT scans of patients with NSCLC undergoing ICIs; in this study, changes in the intra-tumoral and peritumoral tissue could predict RECIST response (*p* < 0.05) and were associated with OS (HR 1.64, 95% CI 1.22–2.21).

There are some potential limitations associated with our study. First, this was a monocentric retrospective study with a small population (63 patients). While most of these patients were in the D21 group, 10 patients, i.e., 15%, were excluded essentially due to the suboptimal quality of DCE-US examinations. Nonetheless, our results showed a significant association with overall survival at D21 and are consistent with a larger multicenter study that analyzed the same tool (DCE-US) with another therapeutic class. Second, the studied population was heterogeneous: it showcased different cancer types associated with various histological characteristics and a difference in the ICIs used in this study. Third, DCE-US explores and evaluates only one lesion, which may not represent all tumor lesions. Furthermore, we did not compare the ultrasound results with the CT scan, which is still the reference evaluation modality for anti-cancer treatments. The comparison might constitute the topic of further study. Finally, further retrospective studies on a much larger population, or a prospective one, would provide stronger proof of the validity of this biomarker.

## 5. Conclusions

We found that the decrease in perfusion parameters measured by DCE-US was significantly associated with the prolonged survival of patients treated with ICIs. A decrease of more than 45% in AUC at D21 seems a promising biomarker. Additional studies on a larger population would be useful to find a robust threshold to define non-responders earlier after the start of treatment, avoiding a harmful loss of time. Finally, it would be interesting to study perfusion under DCE-US at regular intervals from the first days after the introduction of ICIs in a large cohort to try to find a specific vascular profile according to each type of response to immunotherapy (pseudo-progression, hyper-progressive disease, response, or non-response).

## Figures and Tables

**Figure 1 cancers-14-01337-f001:**
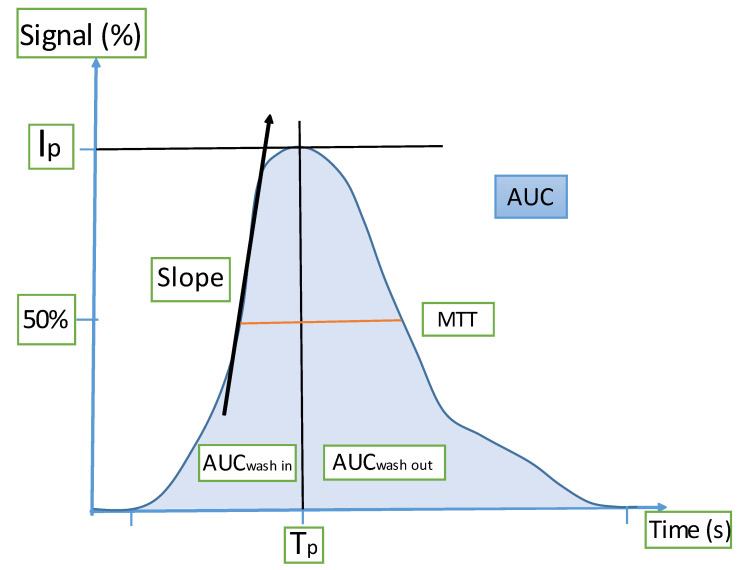
Time-intensity curve modeled from DCE-US. The measured parameters are related to blood volume, blood flow, and blood transit time. (Tp: time to peak; AUC: area under the curve; Ip: peak intensity; MTT: mean transit time).

**Figure 2 cancers-14-01337-f002:**
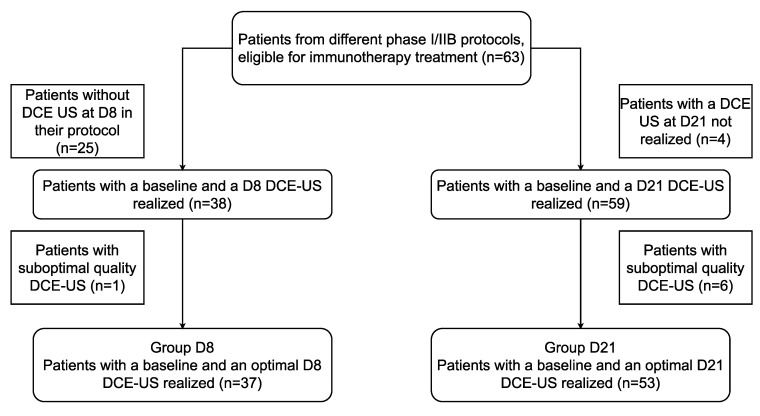
Study flow chart.

**Figure 3 cancers-14-01337-f003:**
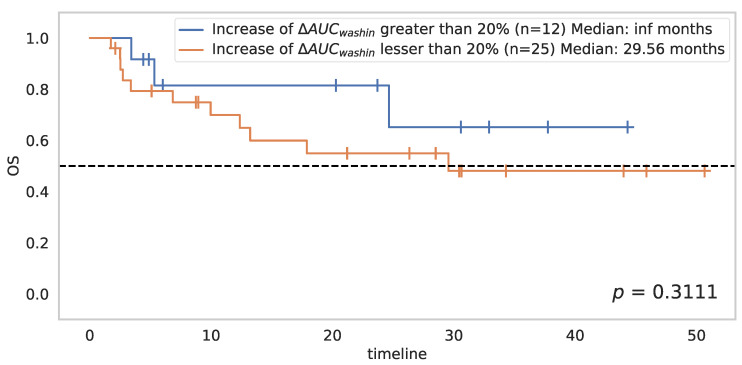
Impact of the overall survival of increase in AUC_wash in_ from baseline to D8 greater than 20%.

**Figure 4 cancers-14-01337-f004:**
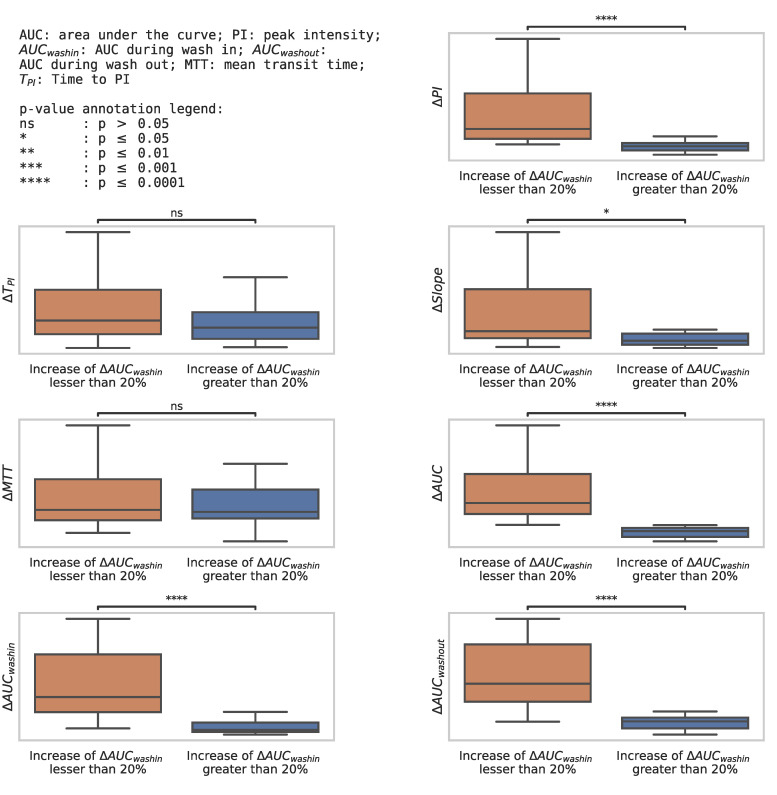
Box plot representing the change in perfusion parameters from baseline to D8 in the group of increase of ΔAUC_wash in_ greater than 20% (=12 patients) vs. in the group of increase of ΔAUC_wash in_ lesser than 20% (=25 patients).

**Figure 5 cancers-14-01337-f005:**
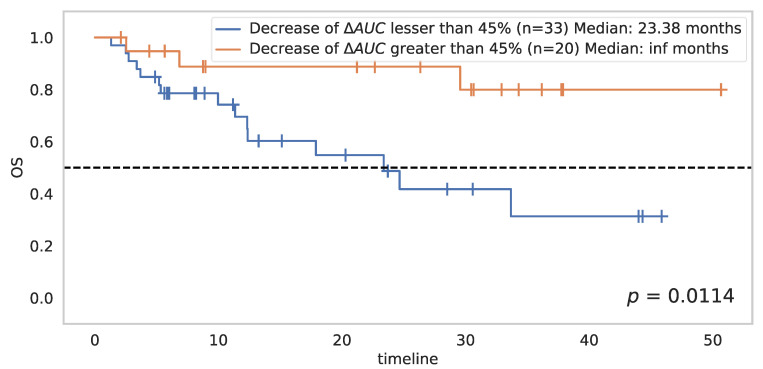
Impact on overall survival of a decrease in AUC from baseline to D21 greater than 45%.

**Figure 6 cancers-14-01337-f006:**
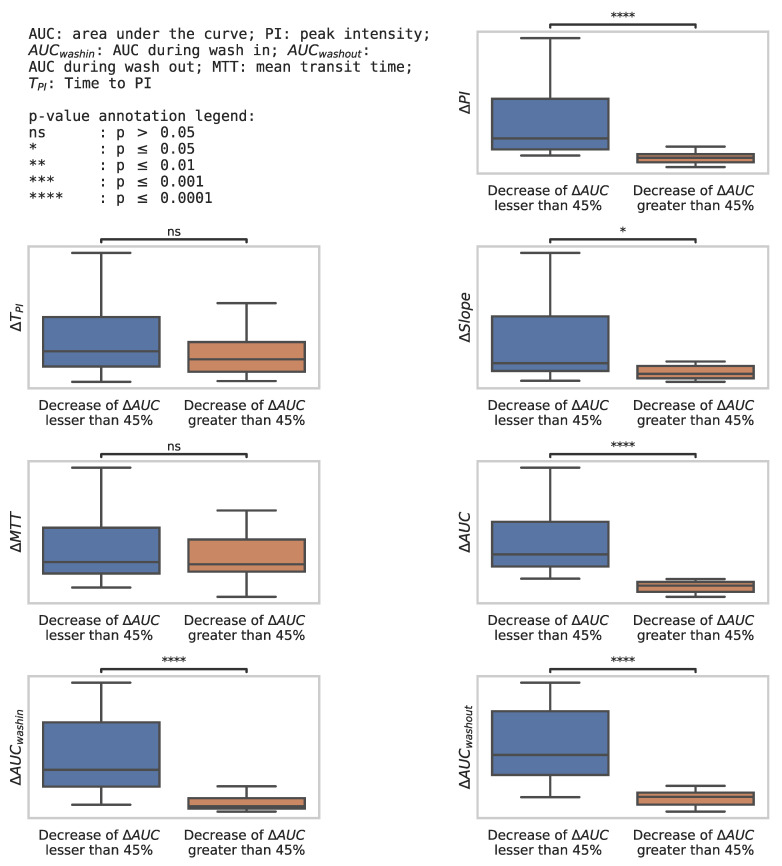
Box plot representing the change in perfusion parameters from baseline to D21 in the group of a decrease in ΔAUC lesser than 45% (*n* = 33 patients) vs. in the group of a decrease in ΔAUC greater than 45% (*n* = 20 patients).

**Figure 7 cancers-14-01337-f007:**
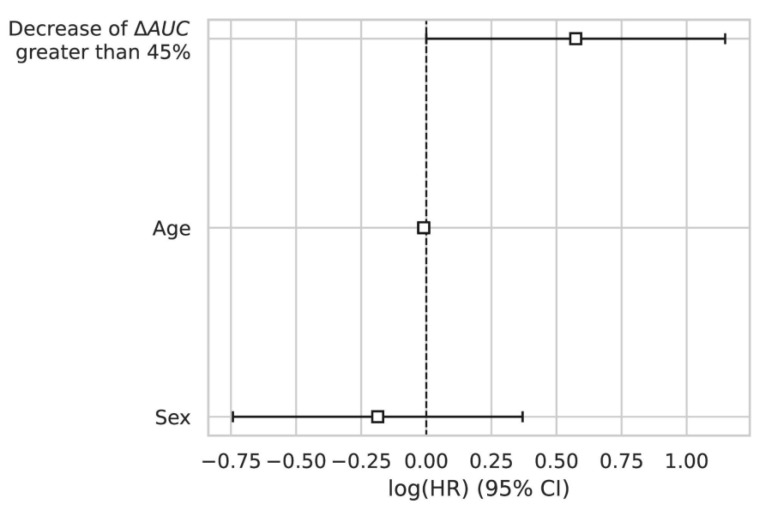
The log hazard ratio of each parameter included in a Cox model. Covariates studied were age, sex, and AUC decrease between baseline and D21 greater than 45%. Only the latter was significantly correlated to survival (HR = 1.75; *p*-value = 0.05).

**Figure 8 cancers-14-01337-f008:**
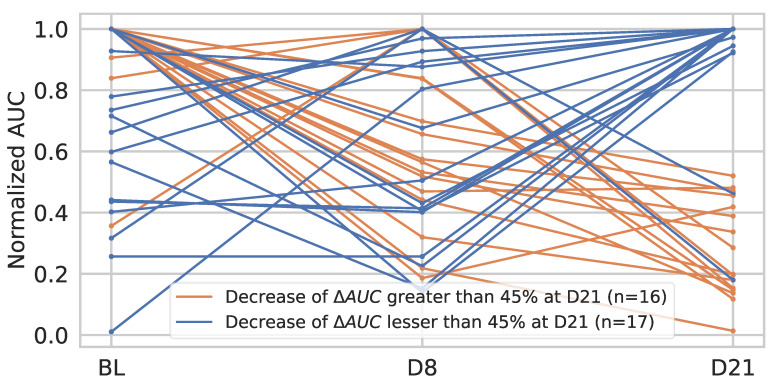
AUC variation from baseline to D8 and from D8 to D21 in the 33 patients with DCE-US at D8 and D21.

**Figure 9 cancers-14-01337-f009:**
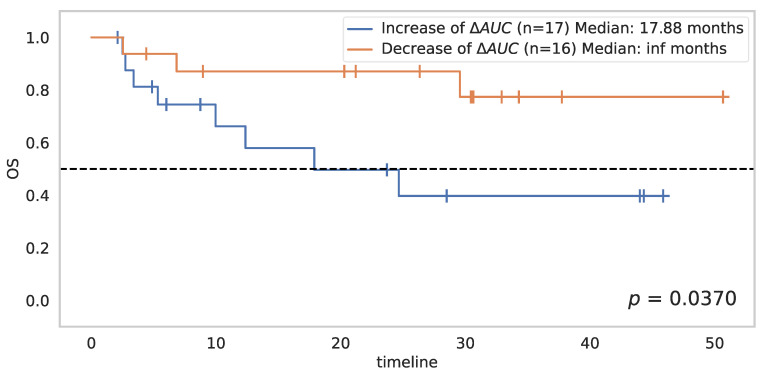
Survival in accordance with AUC variation from D8 to D21 (ΔAUC decrease of any amount) in the 33 patients with DCE-US at D8 and D21.

**Table 1 cancers-14-01337-t001:** Demographics and baseline patient characteristics.

Characteristics	No.	%
Total patients	63	100
Age (Median (IQR)) in years	56 (45–63)	
Male	38	60
Female	25	40
Tumor type		
Melanoma	22	35
Sarcoma	16	25
Colorectal cancer	10	16
Kidney cancer	11	18
Hepatocellular cancer	4	6
Treatment		
Atezolizumab	25	40
Nivolumab	22	35
Pembrolizumab	16	25

**Table 2 cancers-14-01337-t002:** Significance level (*p*-value) of the association between changes in perfusion parameters values between baseline and D8 and overall survival.

ΔD8/BASELINE	*p*-Value
ΔAUC	0.2529
ΔPI	0.4810
ΔAUC_wash in_	0.0593
ΔAUC_wash out_	0.2529
ΔSlope	0.2885
ΔMTT	0.4732
ΔT_PI_	0.0892

**Table 3 cancers-14-01337-t003:** Significance level (*p*-value) of the association between changes in perfusion parameters values between baseline and D21 and overall survival.

ΔD21/BASELINE	*p*-Value
ΔAUC	0.0028
ΔPI	0.0058
ΔAUC_wash in_	0.0294
ΔAUC_wash out_	0.0081
ΔSlope	0.0592
ΔMTT	0.1387
ΔT_PI_	0.2876

## Data Availability

The data presented in this study are available on request from the corresponding author and Nathalie LASSAU.

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
