# Peer review of "Prediction of Early Response to Immunotherapy: DCE-US as a New Biomarker"

_cancers, 2022, doi:10.3390/cancers14051337_

Round 1
Reviewer 1 Report
This manuscript is well written, describing both the novel aspects of the study and inherent limitations. Used in conjunction with CT and other clinical assessments, this may lead to a much-needed protocol for earlier detection of immunotherapy response.
Author Response
We thank the reviewer for their time spent carefully reviewing the manuscript, and we appreciate their positive feedback regarding the information presented in this paper.
Reviewer 2 Report
The authors should present the detailed parameters of DCE-US, made publicly available.
Minor issues:
Figure 1.
Figure 6, no title in Y-axis.
Line272: the P-value is wrong.
Author Response
We thank the reviewer for their time spent carefully reviewing the manuscript. We made sure that each comment has been addressed, and hope that our responses satisfy their remarks.
- The authors should present the detailed parameters of DCE-US, made publicly available.
We initially thought that detailing the parameters of DCE-US was not necessary, since they have been explicitly described in the referred paper (17). We now agree that it is important for readers not very familiar with DCE-US to detail these parameters in order to understand the current study. As requested, we added information about the extracted DCE-US parameters, and a figure to illustrate their meanings in Methods DCE-US Technique.
Minor issues:
- Figure 1.
We designed Figure 1 anew in order to make it simpler and more understandable.
- Figure 6, no title in Y-axis.
The plot represents DCE-US AUC perfusion vs. time. Since the raw values differ substantially between patients, we have normalized them for each patient, in order to represent the relative change of these parameters. This is performed by dividing each patient’s data by the maximum value of AUC of the three time points. The data is therefore dimensionless.
We have added the title “Normalized AUC”.
- Line 272: the P-value is wrong.
We corrected the p-value to match the one presented in the cited paper.
Reviewer 3 Report
In their current study authors aimed to develop a tool to assess the early response to ICIs using data from DCE-US scans. The findings are interesting. Few suggestions to the authors are listed below.
- Authors could elaborate the statistical analysis plan. For example, it is not clear how the cut-off points were identified (e.g. page 5, line, 171). Secondly, from the presentation it appears that authors performed multiple tests of significance and found a marker by chance. Clear stepwise description of the statistical plan that was to be implemented during the study would help address the concern.
- Authors could also explain what the difference in baseline and day 8/21 would mean (e.g. increased/decreased blood flow or less/more tumors) within the methods section. Readers would appreciate description of how the changes in parameters are physiologically related to response to ICI.
- Authors could use univariate and multivariate regression (age, gender, tumor stage as covariates) analyses to further confirm their findings.
- Sample size is small (not relatively small). Authors had data from only 53 patients for day 21 and 37 patients for day 8, which can only help in giving an initial signal. For example, current study identified 45% at day 21 as a cut-off whereas previous study identified it as 40% change at 1 mo, which are not the same. Further studies in larger cohorts are needed to validate the marker. Authors could acknowledge the point.
- Authors also could acknowledge that a prospectively designed study would be needed to confirm the marker.
Minor
- Page 2, line 66 please correct ‘efficacity’
- Authors could use the help of professional English writer to address grammatical errors (e.g. page 11, “in a study assessing DCE-MRI perfusion to predicts pseudoprogression in metastatic melanoma”)
- Authors could cite multiple reviews on immunotherapy such as Makaremi et al (https://doi.org/10.3390/biomedicines9091075), Lemaire et al (https://doi.org/10.1186/s13046-021-02111-5), Relecom et al (https://doi.org/10.1186/s13046-021-01872-3), and Yu et al (https://doi.org/10.3390/biomedicines9111702) to give readers additional sources of information on the topic.
Author Response
We thank the reviewer for their time spent carefully reviewing the manuscript. We made sure that each comment has been addressed, and hope that our responses satisfy their remarks. The reviewer's observations are in Black, our responses are in Red.
- Authors could elaborate the statistical analysis plan. For example, it is not clear how the cut-off points were identified (e.g. page 5, line, 171). Secondly, from the presentation, it appears that authors performed multiple tests of significance and found a marker by chance. Clear stepwise description of the statistical plan that was to be implemented during the study would help address the concern.
We did not find the marker by chance, but by a predictive study motivated by previous work.
We convey as much as we can of the statistical plan behind this study in the Analysis and Discussion sections, summarized here:
- To select the cut-off points, we used maximally selected ranking tests, which is a well-established methodology (B Lausen, M Schumacher - Biometrics, 1992)
- A previous study on anti-angiogenic agents showed that a 40% decrease in AUC at one month was significantly correlated to OS. In the present study, with immunotherapy agents, we found that a 45% decrease at D21 is also significant, and is obtained quicker. The two metrics measure the same phenomenon: a decrease in AUC over time. Only the magnitude and time interval of the decrease are different.
- We have also studied other quantitative parameters measurable with DCE-US between baseline, D8 and D21. This study shows that no other variation is as significant as the variations of AUC detailed in the article.
- The 20% threshold chosen for the variation between baseline and D8 was obtained using maximally selected rank statistics, with the added constraint that each subgroup should have a number of patients greater or equal to 30% of the total. The motivation behind this study was to find an even faster way of evaluating patients. Unfortunately, it turns out to not be significant, so we did not explore it further. Nonetheless, we still wish to mention it as it shows a weak signal that may be explored in further studies, particularly with a larger prospective cohort.
- The box plots presented emphasize that the distributions are statistically and significantly different.
- Authors could also explain what the difference in baseline and day 8/21 would mean (e.g. increased/decreased blood flow or less/more tumors) within the methods section. Readers would appreciate description of how the changes in parameters are physiologically related to response to ICI.
We have added more information about our interpretation concerning the results in the discussion section (end of second paragraph of Discussion), and provided a meaning to the parameters as requested by R3 and another reviewer (in methodology section DCE-US technique and Quantification).
Our results showed that an increase in AUC between baseline and D8 was weakly associated with better survival, while a decrease in AUC between baseline and D21 and between D8 and D21 was significantly associated with better survival. This means that following the introduction of immunotherapy, an increase in perfusion between baseline and D8 and a decrease in perfusion between D8 and D21 should be predictive of a good response to the treatment.
To explain this phenomenon physiologically, we could rely on the nature of the immune response; the goal of immunotherapy is to restore the ability of the immune system to fight against tumor cells. When the immune system reacts to aggression (in this case against tumor cells), a pro-inflammatory environment is created (via a variety of cytokines).
One of the aspects of this environment is the initiation of neo-angiogenesis that allows an influx of immune cells to eliminate the tumor cells. This leads to the destruction of tumor vasculature along with tumor cells, resulting in necrosis, hence explaining the decrease of perfusion in case of response.
In summary: the increase in perfusion at D8 would indicate initiation of neo-angiogenesis, whereas a decrease in perfusion at D21 would indicate necrosis and the destruction of tumor vasculature since the tumor cells are eliminated. Both would be an indication of a good response to the treatment.
- Authors could use univariate and multivariate regression (age, gender, tumor stage as covariates) analyses to further confirm their findings.
Since all the patients belong to phase I/IIB protocol studies, the tumor stage is generally III+. Unfortunately, we did not have access to that information during our study. We performed a multivariate regression using a Cox model on the parameters using age and gender as covariates, and the results came out to be significant only for the AUC decrease greater than 45% in Group D21 (HR=1.75, p-value=0.05). The age and gender were not significantly correlated to survival (HR=0.99,p=0.51; HR=0.83,p=0.37 respectively). Here’s a plot with log(HR) of the parameters in the case where AUC variation came out to be significant. We have added the plot and an explanation of this result in the results sections Baseline-D21 (Supplementary Figure 2).
- Sample size is small (not relatively small). Authors had data from only 53 patients for day 21 and 37 patients for day 8, which can only help in giving an initial signal. For example, current study identified 45% at day 21 as a cut-off whereas previous study identified it as 40% change at 1 mo, which are not the same. Further studies in larger cohorts are needed to validate the marker. Authors could acknowledge the point.
We agree with the reviewer’s observation and have modified the text to put some emphasis on this point:
- We had previously mentioned in the conclusion that: “Additional studies would be useful to find a robust threshold to define non-responders earlier after the start of treatment, avoiding a harmful loss of time.” We believe that it does acknowledge the fact that our study is not sufficient to validate the biomarker. We modified the sentence like so: “Additional studies on a larger population would be useful to find a robust threshold to define non-responders earlier after the start of treatment, avoiding a harmful loss of time”.
- We added a sentence to put an emphasis on the fact that the cut-offs are not necessarily the same between the cited paper and our result, in the discussion section (second paragraph). The sentence is: “Although the cutoffs are not identical, our study was highly motivated by that result, since the studied criterion (Decrease of AUC) as a relevant marker for overall survival is the same.”
- We removed the “relatively” in “relatively small”. We do not pretend that our study validates the biomarker, and do hope not to mislead the reader into thinking that.
- We added a small sentence in the last paragraph of the discussion section, stating the need for a bigger cohort as one of the limits of our study. “Finally, further retrospective studies on a much larger population, or a prospective one, would provide stronger proof of the validity of this biomarker.”
Of course, our sample is too small to validate a precise cut-off or to say that DCE-US could replace CT scan. Our goal was primarily to show if ultrasound quantitative markers could be used to evaluate the therapeutic response of immunotherapy. The most important result of our study is to show that the decrease in tumor perfusion (which is quickly and easily measured by DCE-US) is a predictive factor of good therapeutic response, as exhibited by a significantly better OS. Further studies with larger cohorts are indeed needed to validate the marker.
- Authors also could acknowledge that a prospectively designed study would be needed to confirm the marker.
We agree that a prospective design study is needed to confirm the marker, and hope that our study may encourage other researchers or oncologists to design one.
Minor
- Page 2, line 66 please correct ‘efficacity’
Thank you, the correction was made.
- Authors could use the help of professional English writer to address grammatical errors (e.g. page 11, “in a study assessing DCE-MRI perfusion to predicts pseudoprogression in metastatic melanoma”)
Two of our co-authors are native English speakers and are also academics. We also requested other lab members to read and correct any spelling mistakes, and we hope to have corrected everything.
- Authors could cite multiple reviews on immunotherapy such as Makaremi et al (https://doi.org/10.3390/biomedicines9091075), Lemaire et al (https://doi.org/10.1186/s13046-021-02111-5
- ), Relecom et al (https://doi.org/10.1186/s13046-021-01872-3), and Yu et al (https://doi.org/10.3390/biomedicines9111702) to give readers additional sources of information on the topic.
We thank you for pointing us to these papers. We have added these references in the introduction and the discussion.
Round 2
Reviewer 3 Report
No comments